# Low Expectancy of Conversion Surgery with R0 Resection in Patients with CEA > 5.0 ng/mL at the Initial RECIST Evaluation for Metastatic Gastric Cancer

**DOI:** 10.3390/cancers15215197

**Published:** 2023-10-29

**Authors:** Koki Nakanishi, Chie Tanaka, Mitsuro Kanda, Kazushi Miyata, Kazuhiro Furukawa, Osamu Maeda, Dai Shimizu, Shizuki Sugita, Naomi Kakushima, Satoshi Furune, Hiroki Kawashima, Yuichi Ando, Tomoki Ebata, Yasuhiro Kodera

**Affiliations:** 1Department of Gastroenterological Surgery, Nagoya University Graduate School of Medicine, Nagoya 466-8550, Japan; 2Division of Surgical Oncology, Department of Surgery, Nagoya University Graduate School of Medicine, Nagoya 466-8550, Japan; 3Department of Gastroenterology and Hepatology, Nagoya University Graduate School of Medicine, Nagoya 466-8550, Japan; 4Department of Clinical Oncology and Chemotherapy, Nagoya University Hospital, Nagoya 466-8560, Japan

**Keywords:** metastatic gastric cancer, conversion surgery, chemotherapy, tumor marker, predictor

## Abstract

**Simple Summary:**

This study aimed to identify early predictive factors associated with the success of conversion surgery with R0 resection in patients with metastatic gastric cancer who underwent systemic chemotherapy. Patients with carcinoembryonic antigen > 5.0 ng/mL at the initial Response Evaluation Criteria in Solid Tumors evaluation showed less expectancy of undergoing conversion surgery with R0 resection.

**Abstract:**

This retrospective study examined early the predictive factors for successful conversion surgery (CS) with R0 resection in patients with metastatic gastric cancer (MGC) who underwent systemic chemotherapy. This study included 204 patients diagnosed with metastatic gastric adenocarcinoma, who received chemotherapy between 2009 and 2019. Of these patients, 31 (15%) underwent CS with R0 resection. The incidence of CS with R0 resection was not affected by the volume of metastatic lesions or the presence of peritoneal metastasis. The overall survival time of the CS with R0 resection group was significantly longer than that of the non-CS group (hazard ratio, 0.12; 95% confidence interval, 0.07–0.23; *p* < 0.0001), with a 5 year overall survival rate of 50.2%. Multivariate analysis of 150 patients, excluding those with disease progression until the initial Response Evaluation Criteria in Solid Tumors (RECIST) evaluation, showed that carcinoembryonic antigen > 5.0 ng/mL at the initial RECIST evaluation was an independent, significant, and unfavorable predictor of CS with R0 resection (odds ratio, 0.21; *p* = 0.0108), whereas systemic chemotherapy with trastuzumab for HER2-positive cancer was a favorable factor (odds ratio, 4.20; *p* = 0.0119). Monitoring serum carcinoembryonic antigen levels during chemotherapy may be a useful predictor of the CS implementation in patients with MGC.

## 1. Introduction

Systemic chemotherapy is the standard treatment to palliate symptoms and prolong survival in metastatic gastric cancer (MGC) [1]. Although the median survival time remains unsatisfactory, between 6 and 14 months, complete or near-complete response (CR or near-CR) of the metastatic lesions sometimes occurs. This renders the primary lesion ± the remains of the metastases resectable [2]. This surgery has recently been referred to as conversion surgery (CS), and several retrospective studies have shown its favorable outcomes [3,4,5,6,7,8,9,10,11]. Nevertheless, the clinical significance of CS remains unclear, owing to the lack of prospective clinical trials.

In general, CS should only be performed on patients with excellent responses to pharmacotherapy, with hopes for R0 resection, and who are physically and mentally prepared to undergo major surgery. However, owing to the diversity in the type and volume of metastatic lesions in more complex conditions and the frequent presence of lesions undetected by conventional imaging studies, it is challenging to design an appropriate clinical trial to explore CS for MGC. Thus, before conducting such a trial, an objective and simple index should be developed to select suitable candidates for CS. Furthermore, it is known that tumor response evaluated by computed tomography (CT) imaging and a marked decrease in serum tumor marker values during chemotherapy correlate with better prognosis in various malignancies, including MGC [12,13,14,15,16,17].

In this study, we aimed to retrospectively analyze patients diagnosed with MGC and who were treated with systemic chemotherapy at our institution in order to identify the early predictive factors associated with the success of CS with R0 resection.

## 2. Materials and Methods

### 2.1. Patients

Consecutive patients diagnosed with metastatic gastric adenocarcinoma who received systemic chemotherapy at Nagoya University Hospital between January 2009 and December 2019 were included in this study. We classified them into two groups based on whether they underwent CS (CS and non-CS groups). In this study, patients with technically resectable metastases at the initial diagnosis, or synchronous malignancies, were excluded. Additionally, patients with esophagogastric junction cancer were also excluded. Using previous biological categories (Yoshida classification) [18], technically resectable metastasis was defined and classified as one of the following: (1) ≤2 metastatic nodules in the liver with ≤5 cm diameter (maximum), (2) positive cytology status (CY1), or (3) suspected para-aortic lymph node metastases in the region between the celiac axis and inferior mesenteric artery (no. 16a2 and/or 16b1). Tumor staging and histopathological grading were performed according to the 15th edition of the Japanese Classification of Gastric Carcinoma [19]. Distant metastases, other than technically resectable metastases as defined by the Yoshida classification, were defined as non-curable factors. Hematologic metastasis was defined as metastasis to the bone, liver, lung, and other organs, and lymphogenous metastasis was defined as distant lymph node metastasis. The flow diagram of the patients enrolled in this study is shown in Figure 1.

### 2.2. Chemotherapy, Surgery, and Postoperative Management

Chemotherapy was administered according to the latest Japanese gastric cancer guidelines or clinical trial regimens at that time. Tumor responses were evaluated according to the Response Evaluation Criteria in Solid Tumors (RECIST) version 1.1 [20]. Serum levels of carbohydrate antigen 19-9 (CA19-9) and carcinoembryonic antigen (CEA) were measured in all patients before and during chemotherapy (every 6 ± 2 weeks). The cutoff values for CEA and CA19-9 were set at the upper limit of the normal value for each tumor marker at our institution: CEA, 5.0 ng/mL and CA19-9, 37 U/mL. CS was defined as curative-intent gastrectomy with ≥D1 lymph node dissection and was considered when R0 resection was possible. This study did not have a set protocol for CS implementation; however, it was undertaken in the following circumstances: (1) for peritoneal dissemination, if the lavage cytology showed negative results, and a representative peritoneal nodule was confirmed to be pathologically negative at the staging laparoscopy, with no evidence of other distant metastases; (2) for liver metastasis, when a CR or a reduction to 1 or 2 nodules with a maximum diameter of <5 cm was ensured; and (3) for lymph nodes or other distant sites, when a CR was confirmed in the distant metastasis. Lesions in which CR was achieved did not necessarily require surgical removal.

The Clavien–Dindo classification was employed to comprehensively evaluate postoperative complications [21,22], such as the clinically relevant grade ≥ II complications. The pathological response of the primary tumor after preoperative chemotherapy was assessed according to the 5th edition of the Japanese Classification of Gastric Carcinoma [19]. The regimen and duration of postoperative chemotherapy were at the physician’s discretion, considering the surgical and pathological findings, as well as the patient’s condition.

### 2.3. Statistical Analysis

The correlation between each group was analyzed using the chi-squared test or Fisher’s exact test for categorical variables and the Mann–Whitney U test for continuous variables, where appropriate. Survival rates were estimated using the Kaplan–Meier method, and the overall differences between survival curves were compared using the Cox proportional hazards model. Overall survival (OS) was calculated as the time between the start of chemotherapy and death/last follow-up date. Recurrence-free survival (RFS) was defined as the time between surgery and the first evidence of disease recurrence or progression and was calculated for patients who underwent R0 resection. Multivariable logistic analysis was performed to identify important factors for CS with R0 resection. JMP version 15 (SAS Institute Inc., Cary, NC, USA) was used for all of the analyses. Statistical significance was set at *p* < 0.05.

## 3. Results

### 3.1. Patients’ Characteristics

In this study, 15 patients with technically resectable metastases were excluded (2 cases of ≤2 metastatic nodules in the liver with ≤5 cm maximum diameter, 9 cases of positive cytology, and 9 cases of para-aortic lymph node metastases confined to 16a2 and/or 16b1). A total of 204 patients were included in this analysis, of whom 39 underwent CS (conversion rate, 19%). Table 1 summarizes the patient characteristics of the two groups. Patients in the CS group had significantly better ECOG-PS scores, were female, and were younger than those in the non-CS group. There were no significant differences between the two groups in terms of histological type, distant metastasis site, or number of non-curable factors. However, the proportion of patients who received systemic chemotherapy combined with intraperitoneal administration and trastuzumab was significantly higher in the CS group than that in the non-CS group.

### 3.2. CS

Table 2 summarizes the operative variables in the CS group. R0 resection was achieved in 31 patients (conversion surgery with R0 resection rate, 15%). At the time of CS, CR, partial response (PR), stable disease (SD), and non-CR/non-PD with chemotherapy were established in 1 (2.6%), 26 (67%),4 (10%), and 8 (20%) patients, respectively. The percentage of CS cases with R0 resection was 12% for Yoshida classification category 2, 17% for category 3, and 16% for category 4, with no significant variation between the categories (*p* = 0.6425). The surgical procedures for combined resection included the following: three partial hepatectomies, one lateral segmentectomy, and two distal pancreatectomies. Postoperative complications occurred in 8 (21%) patients. Postoperative chemotherapy was administered to 35 (90%) patients. The pathological response assessment of primary tumors showed that 10 (26%) patients had a grade 2a, 2 (5%) patients had a grade 2b, and 5 (13%) patients had a grade 3 response.

### 3.3. Survival Outcomes

The median follow-up period was 8.7 months (interquartile range (IQR), 4.3–14.4 months) in the non-CS group and 39.8 months (IQR, 17.3–56.6 months) in the CS group. The median survival times (MST) were 77.3 months in the CS with R0 resection group, 19.3 months in the R1/2 resection group, and 9.4 months in the non-CS group. The hazard ratios for OS in the R0 resection group were 0.12 (95% CI, 0.07–0.23; *p* < 0.0001) compared with the non-CS group, while in the R1/2 resection group, it was 0.49 (95% CI, 0.24–1.01; *p* = 0.0537). The hazard ratios for OS compared with the R1/2 group were 0.25 (95% CI, 0.11–0.63; *p* = 0.0029) in the R0 resection group (Figure 2). This trend was the same for all of the Yoshida classification categories (Figure 3).

Fifteen patients with technically resectable metastases underwent induction chemotherapy. Of these, 14 patients subsequently underwent radical surgery, with 2 achieving an R1 resection and 12 achieving a R0 resection. The 5 year OS rate for R0 resection was 62.5%.

The RFS was analyzed in 31 patients who underwent CS with R0 resection. The median RFS was 29.4 months (IQR, 5.7–49.9 months), and the 3-year RFS rate was 43.8%. Among these, patients who received postoperative chemotherapy had significantly longer RFS times than those who did not (HR, 0.22; 95% CI, 0.06–0.83; *p* = 0.0250, Figure 4a). Patients with pathological response grade ≥ 2 had a longer RFS times, but the difference was not significant (HR, 0.49; 95% CI, 0.19–1.23; *p* = 0.1301, Figure 4b).

### 3.4. Early Predictive Factors for CS with R0 Resection

We analyzed the factors that influenced the early prediction of CS until the initial RECIST evaluation. This analysis was conducted on a subset of 150 patients, excluding those with progressive disease or who had discontinued chemotherapy until the initial RECIST evaluation. Patient and tumor characteristics, initial RECIST evaluation, and serum tumor markers at the initial RECIST evaluation were considered to be variables for the early prediction of CS with R0 resection (Table 3). Univariate analysis revealed that the following predictive factors were significant when implementing CS with R0 resection: CEA > 5.0 ng/mL at the initial RECIST evaluation and chemotherapy combined with trastuzumab administration. Multivariate analysis revealed that CEA > 5.0 ng/mL at the initial RECIST evaluation was an independent and unfavorable factor when implementing CS with R0 resection (odds ratio (OR), 0.21; 95% CI, 0.07–0.70; *p* = 0.0108), whereas chemotherapy with trastuzumab administration was an independent and favorable factor (OR, 4.20; 95% CI, 1.37–12.9; *p* = 0.0119).

### 3.5. Relationship between Changes in CEA Levels and CS with R0 Resection

Figure 5 shows the association between changes in CEA levels from baseline to the initial RECIST evaluation and CS with R0 resection. Among the 64 patients with CEA levels > 5.0 ng/mL at baseline, 16 (25.0%) exhibited a decrease in CEA levels to ≤5.0 ng/mL at the initial RECIST evaluation. In contrast, among the 86 patients with CEA levels ≤ 5.0 ng/mL at baseline, 4 (4.7%) had an increase in CEA levels to >5.0 ng/mL at the initial RECIST evaluation, and none of these patients achieved CS with R0 resection. Of the patients with CEA levels ≤ 5.0 ng/mL at the initial RECIST evaluation (*n* = 98), 27 (28%) achieved CS with R0 resection. In contrast, only 4 (7.7%) patients with CEA levels > 5.0 ng/mL at the initial RECIST evaluation (*n* = 52) achieved CS with R0 resection, which was significantly less frequent (*p* = 0.0043).

## 4. Discussion

We retrospectively analyzed patients diagnosed with MGC who were treated with systemic chemotherapy at our institution in order to identify the early predictive factors associated with the success of CS with R0 resection. The incidence of CS with R0 resection was 15%, and the patients who underwent the procedure had a better prognosis. However, patients with CEA > 5.0 ng/mL at the initial RECIST evaluation had a low indication for surgery.

The reported incidence of CS varies from 17.7% to 34.9% in large-scale retrospective studies [8,9,10,11]. This is slightly higher than our findings, potentially because patients with technically resectable metastases (Yoshida classification category 1) were excluded from our study. Chemotherapy for such patients could be regarded as induction chemotherapy, and we believe that our eligibility criteria were reasonable. Additionally, there may have been patients in our series who missed the timing for the implementation of CS because of our cautious approach toward the procedure. The incidence of CS observed in prospective clinical trials of chemotherapy for advanced/metastatic cancer is much lower, at 4.2% and 6%, respectively [23,24]. This may be because our series included several HER2-positive patients treated with trastuzumab-containing regimens and those with peritoneal disease who were treated with intraperitoneal chemotherapy. Moreover, all investigators who participated in the previous clinical trials were keen on conducting CS. We are currently conducting a multi-institutional prospective observational study, using the same inclusion and exclusion criteria, to evaluate the clinical significance and status of CS (UMIN000042244).

We recently reported that postoperative complications, massive intraoperative blood loss, and delayed postoperative adjuvant chemotherapy adversely affect the prognosis of patients with gastric cancer [25,26,27]. Thus, our CS strategy avoided extended surgery (e.g., extended lymph node dissection, pancreaticoduodenectomy, and Appleby’s procedure). We also made efforts in this series to continue chemotherapy after surgery. Our subgroup analysis showed that patients who received postoperative chemotherapy had better prognoses than those who did not.

Our study suggests that changes in tumor marker levels could serve as an early indicator in patients with gastric cancer. Tumor shrinkage is an important indicator when considering whether CS is indicated in daily practice. However, relying solely on RECIST evaluation may not suffice, as conventional imaging studies may fail to detect lesions, such as those associated with peritoneal dissemination, which is often associated with gastric cancer. All cases of R1/2 resection presented with peritoneal dissemination at initial diagnosis. Furthermore, CS requires a CR or near-CR of the metastatic lesions. In this study, CEA > 5.0 ng/mL at the initial RECIST evaluation represented an independent and unfavorable factor for performing CS with R0 resection, whereas the RECIST assessment was not a significant predictor. We speculate that the elevated CEA level more sensitively reflects the presence of metastatic lesions than CT imaging assessments. If the CEA level decreases from baseline, tumor shrinkage may be expected. However, the presence of CEA > 5.0 ng/mL at the initial RECIST evaluation indicates that the cases with CEA > 5.0 ng/mL did not show the marked shrinkage necessary to implement CS, while those with CEA ≤ 5.0 ng/mL had tumor progression. Moreover, CEA trends offer an objective and less invasive method of assessment. However, the wide range of CEA levels (>5.0 ng/mL) may not apply to all cases; the optimal timing of CEA measurement for predicting later CS was unclear in this study. Moreover, the usefulness of other tumor markers, such as MRI and PET/CT, as predictors for implementing CS has not been evaluated. It is necessary to assess this correlation using studies with larger sample sizes. We plan to investigate the relationship between changes in tumor markers and the indication for CS in our ongoing prospective observational study.

Many practice-changing clinical trials have been reported in the literature, and more active agents have become increasingly available in clinical practice for the past decade. Accordingly, based on the results of the ToGA study [28], systemic chemotherapy combined with trastuzumab, the standard treatment for HER2-positive MGC, successfully predicted CS in our study. In combination with chemotherapy, nivolumab has recently become available for the treatment of HER2-negative MGC [29,30], which is expected to have a higher tumor response and may also be an important predictor of CS. However, these patients were not included in this study. Moreover, several patients received systemic chemotherapy combined with intraperitoneal administration in the clinical trial setting; this factor may have impacted a subset of patients with peritoneal disease. A future increase in biomarker-driven treatments may lead to more opportunities for performing CS with R0 resection. At that time, the monitoring of CEA levels during chemotherapy might become increasingly important as these become candidates for CS implementation.

Our study has several limitations. First, this was a single-center, nonrandomized, retrospective study. Second, CS was conducted at the physician’s discretion, and the criteria for implementing the procedure were not fixed. Third, as mentioned above, the availability of active agents in clinical practice has dramatically changed from that at the time of our study period. Fourth, we included patients who underwent clinical trials, and not all chemotherapy regimens were selected based on the Japanese gastric cancer guidelines. Notably, several patients received systemic chemotherapy combined with intraperitoneal administration in a clinical trial setting; this factor may have affected a subset of patients with peritoneal disease. Fifth, tumor markers do not always reflect the total tumor volume or the tumor response to chemotherapy. Sixth, complete metastasis resections were not performed, and the true achievement of R0 resection relies on clinical and imaging assessments. Seventh, although we performed staging laparoscopy before CS in patients with peritoneal dissemination, we still encountered a significant number of R1/2 resection cases. Finally, data regarding good responders to chemotherapy without CS were lacking, and the MST of CS with R1/2 had longer survival times than those in the non-CS group. However, it is important to note that R1/2 resection is considered incomplete and may not provide the same benefits as that of R0 resection in terms of long-term survival. Additionally, patients who undergo R1/2 resection may require further treatment, such as radiation or additional chemotherapy, which can affect overall survival.

## 5. Conclusions

In conclusion, our study highlights the significance of monitoring CEA levels during chemotherapy when predicting the likelihood of achieving CS with R0 resection. Patients with CEA > 5.0 ng/mL at the initial RECIST evaluation might be unfavorable candidates for CS. Monitoring CEA levels during chemotherapy may be a useful predictor of the CS implementation in patients with MGC. Although monitoring CEA trends is an objective and less invasive assessment method, it is essential to emphasize that making treatment decisions solely based on CEA levels is not advisable. Other critical factors, such as the patient’s overall health status, tumor location, and response to treatment, must be carefully considered in the decision-making process. Therefore, the circumstances of each patient should be carefully evaluated in order to determine the most appropriate treatment strategy.

## Figures and Tables

**Figure 1 cancers-15-05197-f001:**
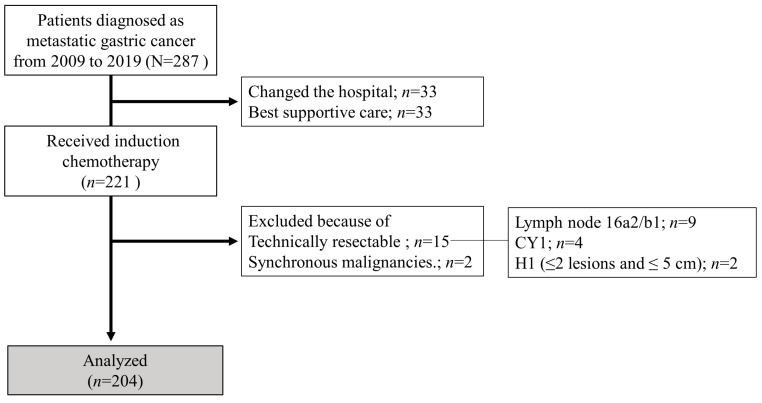
Flowchart of patient enrolment.

**Figure 2 cancers-15-05197-f002:**
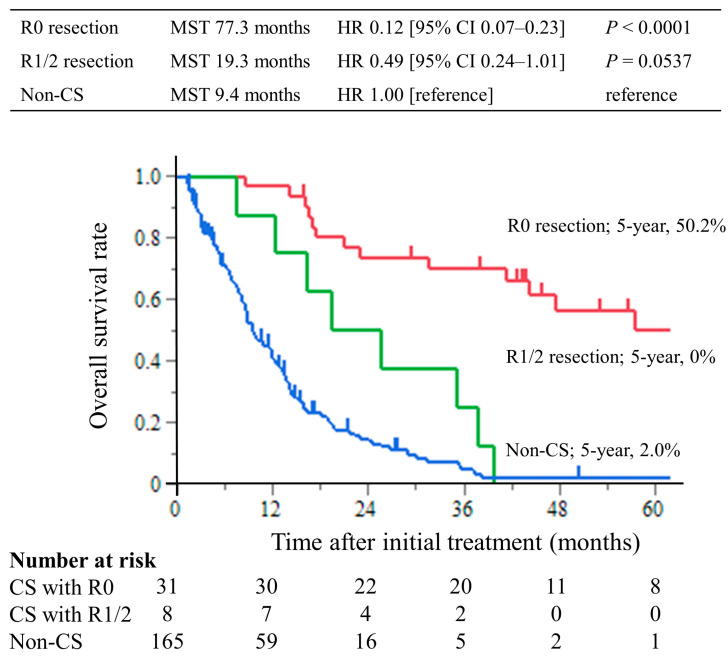
Kaplan–Meier curves of overall survival. The overall survival time of the CS with R0 resection group was significantly longer than that of the non-CS group, with a 5 year overall survival rate of 50.2%.

**Figure 3 cancers-15-05197-f003:**
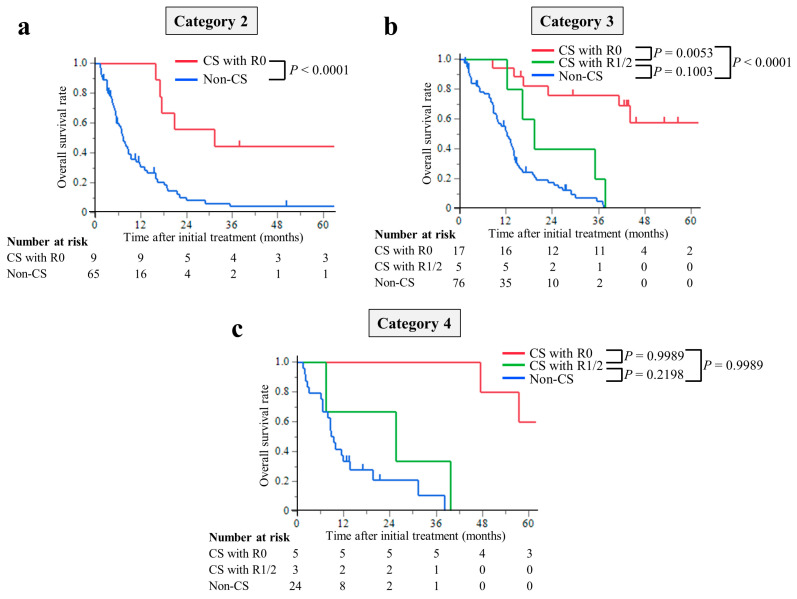
Kaplan–Meier curves for overall survival of patients stratified using the Yoshida classification: (**a**) Category 2, (**b**) Category 3, and (**c**) Category 4. The overall survival time of the CS with R0 resection group was significantly longer than that of the non-CS group in any Yoshida classification category.

**Figure 4 cancers-15-05197-f004:**
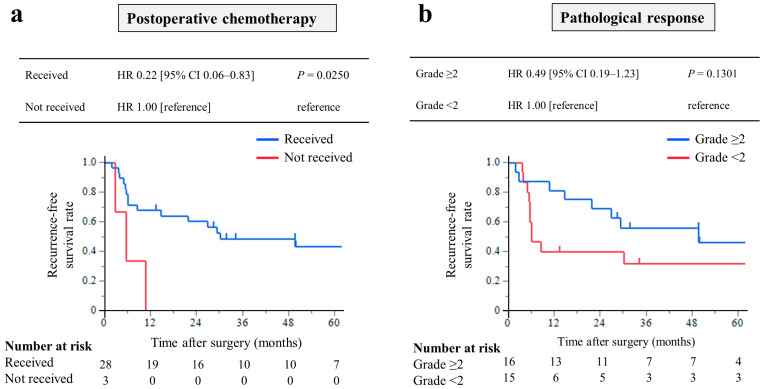
Kaplan–Meier curves of recurrence-free survival of 31 patients who underwent conversion surgery with R0 resection: Analysis of the prognostic effect of (**a**) patients who received postoperative chemotherapy had significantly longer RFS times than those who did not, and (**b**) patients with pathological response grade ≥ 2 had a longer RFS rate, but the difference was not significant.

**Figure 5 cancers-15-05197-f005:**
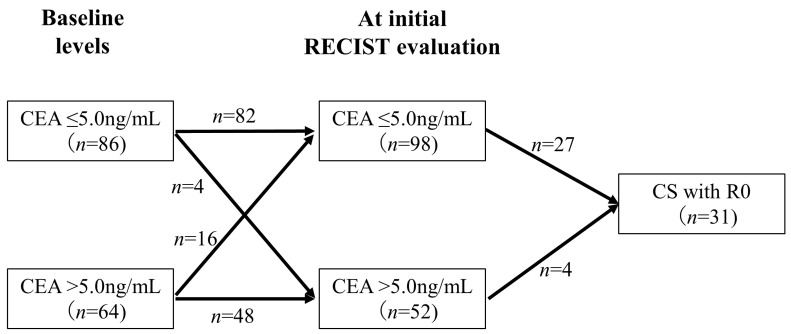
Relationship between changes in CEA levels and CS with R0 resection. This figure shows the association between changes in CEA levels from baseline to the initial RECIST evaluation and CS. Only 7.7% of the patients with CEA levels ≥ 5.0 ng/mL at the initial RECIST evaluation achieved CS with R0 resection, which was significantly less frequent.

**Table 1 cancers-15-05197-t001:** Patients’ characteristics of the study population.

	Non-CS Group *n* = 165	CS Group *n* = 39	*p* Value
Sex			0.0257
Male	108 (65%)	18 (46%)	
Female	57 (35%)	21 (54%)	
Age, years, median (IQR)	66 (59, 73)	61 (49, 68)	0.0027
ECOG-PS			0.0073
0	96 (58%)	32 (79%)	
1	51 (31%)	7 (21%)	
2	18 (11%)	0 (0%)	
ALP, median (IQR)	188 (173, 228)	231 (183, 261)	0.0536
Primary site			0.8904
U	48 (29%)	10 (26%)	
M	66 (40%)	17 (43%)	
L	51 (31%)	12 (31%)	
Macroscopic type			0.1177
Type 1	10 (6%)	1 (3%)	
Type 2	17 (10%)	8 (21%)	
Type 3	77 (47%)	12 (31%)	
Type 4	61 (37%)	18 (46%)	
Histologic type			0.2694
Differentiated	53 (32%)	9 (23%)	
Undifferentiated	112 (68%)	30 (77%)	
Baseline CEA, ng/mL, median (IQR)	3.7 (1.6, 16.8)	2.3 (1.5, 5.6)	0.4986
Baseline CA19-9, IU/mL, median (IQR)	28 (8, 135)	16 (6, 33)	0.3070
cT stage			0.8172
T2	2 (1%)	0 (0%)	
T3	15 (9%)	5 (13%)	
T4a	131 (79%)	30 (77%)	
T4b	17 (10%)	4 (10%)	
cN stage			0.0804
N (−)	30 (18%)	12 (31%)	
N (+)	135 (82%)	27 (69%)	
Yoshida classification			0.1545
2	65 (39%)	9 (23%)	
3	76 (46%)	22 (56%)	
4	24 (15%)	8 (21%)	
Distant metastatic site			
Hematogenous	51 (31%)	8 (21%)	0.1978
Lymphogenous	46 (28%)	13 (33%)	0.4992
Peritoneal dissemination	100 (61%)	30 (77%)	0.0566
Non-curable factors			0.3487
1	123 (75%)	26 (67%)	
2	34 (21%)	9 (23%)	
≥3	8 (5%)	4 (19%)	
Types of chemotherapy			0.0443
S-1	13 (8%)	0 (0%)	
S-1 + paclitaxel	12 (7%)	6 (14%)	
S-1 + oxaliplatin	36 (21%)	6 (16%)	
S-1 + cisplatin	66 (40%)	14 (35%)	
S-1 + docetaxel + cisplatin	2 (1%)	1 (2%)	
Capecitabine + oxaliplatin	3 (2%)	1 (5%)	
Capecitabine + cisplatin	18 (11%)	8 (21%)	
Capecitabine + docetaxel + cisplatin	5 (3%)	3 (7%)	
Others	10 (6%)	0 (0%)	
Combined with trastuzumab administration	19 (12%)	10 (26%)	0.0231
Combined with intraperitoneal administration	30 (18%)	18 (46%)	0.0002

CS, Conversion surgery; IQR, interquartile range; ECOG-PS, Eastern Cooperative Oncology Group Performance Status; CEA, carcinoembryonic antigen; CA19-9, cancer antigen 19-9.

**Table 2 cancers-15-05197-t002:** Surgical, pathological, and postoperative details of patients who underwent conversion surgery.

	*n* = 39
Duration of chemotherapy, months, median (IQR)	7.4 (5.1, 10.3)
RECIST response at conversion surgery	
CR/PR/SD/non-CR/non-PD	1/26/4/8
Type of gastrectomy	
Total gastrectomy/Non-Total gastrectomy	25/14
Surgical approach	
Open/Laparoscopy	38/1
Lymph node dissection	
D1/D1+/D2/D2+	3/2/31/3
Combined resection of other organs	
None/Colon/Liver/Ovary/Pancreas/Spleen	26/3/4/5/2/3
(Partial hepatectomy, 3; Lateral segmentectomy, 1; Distal pancreatectomy, 2)	
Operation time, min	
Mean ± SD ^a^	276 ± 75
Intraoperative blood loss, mL	
Mean ± SD ^a^	789 ± 684
Tumor size, cm	
Mean ± SD ^a^	78 ± 43
ypT stage	
0/1/2/3/4a/4b	5/3/5/10/14/2
ypN stage	
0/1/2/3a/3b	13/7/4/5/10
pStage	
pCR/IA/IB/IIA/IIA/IIIA/IIIB/IIIC/IV	5/3/4/2/2/6/2/2/13
Residual cancer	
R0/R1/R2	31/2/6
Pathological response	
0/1a/1b/2a/2b/3	2/14/6/10/2/5
Postoperative complications ^b^	
II/IIIa/IIIb/IVa/IVb/V	3/4/1/0/0/0
Postoperative chemotherapy	
Absent/Present	4/35

IQR, Interquartile range; CR, complete response; PR, partial response; SD, stable disease; PD progression disease; SD ^a^, standard deviation; ^b^, According to the Clavien–Dindo classification.

**Table 3 cancers-15-05197-t003:** Predictive factors for conversion surgery with R0 resection; univariate and multivariate analyses.

	Univariate			Multivariable		
Variables	OR	95% CI	*p* Value	OR	95% CI	*p* Value
Sex						
Male	1					
Female	1.33	0.61–2.96	0.4705			
Age (years)						
<70	1					
≥70	0.53	0.20–1.41	0.2030			
Yoshida classification						
Category 2	1					
Category 3	1.19	0.48–2.93	0.7079			
Category 4	1.20	0.35–4.11	0.7672			
Non-curable factors						
1	1					
2	1.07	0.41–2.80	0.8838			
≥3	2.49	0.55–11.2	0.2368			
Histological type						
Differentiated	1					
Undifferentiated	1.10	0.46–2.63	0.8247			
Initial RECIST evaluation						
SD, non-CR/non-PD	1					
PR	1.70	0.71–4.06	0.2300			
CEA level at initial RECIST evaluation						
≤5.0 ng/mL	1			1		
>5.0 ng/mL	0.22	0.07–0.67	0.0075	0.21	0.07–0.70	0.0108
CA19-9 level at initial RECIST evaluation						
≤37 U/mL	1					
>37 U/mL	0.59	0.26–1.37	0.2201			
Chemotherapy						
S1/capecitabine + cisplatin/oxaliplatin	1			1		
Combined with intraperitoneal administration	2.48	0.97–6.36	0.0592	1.91	0.68–4.78	0.2342
Combined with Trastuzumab administration	3.84	1.33–11.1	0.0130	4.20	1.37–12.9	0.0119

OR, Odds ratio; CI, confidence interval; RECIST response evaluation criteria in solid tumors; CR, complete response; PR, partial response; SD, stable disease; PD, progressive disease; CEA, carcinoembryonic antigen; CA19-9, cancer antigen 19-9.

## Data Availability

No new data were created or analyzed in this study. Data sharing is not applicable to this article.

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
