# Peer review of "Low Expectancy of Conversion Surgery with R0 Resection in Patients with CEA > 5.0 ng/mL at the Initial RECIST Evaluation for Metastatic Gastric Cancer"

_cancers, 2023, doi:10.3390/cancers15215197_

Round 1
Reviewer 1 Report (Previous Reviewer 1)
Comments and Suggestions for Authors
I confirmed the revised content of the article. I thank you for making the appropriate corrections. The revisions made in response to the suggestions made by other reviewers have been excellent, and the quality of the paper has improved. Although there are differences of opinion, it is worth publishing. I think that this article is worthy of publication with this revision.
Reviewer 2 Report (Previous Reviewer 2)
Comments and Suggestions for Authors
The authors correctly addressed all major concerns raised by the review
Reviewer 3 Report (Previous Reviewer 3)
Comments and Suggestions for Authors
The authors have revised the manuscript from the initial submission. In this revised version, I recommend admission.
This manuscript is a resubmission of an earlier submission. The following is a list of the peer review reports and author responses from that submission.
Round 1
Reviewer 1 Report
Comments and Suggestions for Authors
The authors investigated the factors associated with successful conversion gastrectomy in a retrospective analysis.
Conversion gastrectomy is a topic of gastric cancer treatment today, as chemotherapy has become more effective. The treatment outcomes of this series are good, and the quality of their practice is high. However, the conclusions of this study are commonplace, and the value of this paper is not high.
1. The patients with successful conversion gastrectomy in this study were T-mab cases or intraperitoneal chemotherapy cases. In other words, it can be inferred that the authors' favorable treatment results were derived from the high intensity of treatment. If the intensity of treatment were high, the number of cases in which conversion gastrectomy can be performed would increase, and the CEA in such cases would decrease, reflecting a better response to treatment. In other words, this study states the obvious.
2. How should this result be translated into clinical practice? Is it correct to understand that conversion gastrectomy should be abandoned if the CEA does not decrease even after induction chemotherapy?
3. What clinicians want to know is the content of treatment, the timing of the surgery to achieve R0, and how to plan postoperative chemotherapy. The results of this study do not answer any of these clinical questions.
Comments on the Quality of English LanguageEnglish writing is good.
Reviewer 2 Report
Comments and Suggestions for Authors
This exciting study explores potential predictors of successful conversion to surgery in patients with metastatic gastric cancer after systemic chemotherapy. It is worth mentioning that, to date, the role of curative intent surgery for metastatic gastric cancer remains controversial. The study mainly shows that the OS in patients with metastatic gastric cancer with CS and R0 resection is significantly better than in patients with non-CS. Furthermore, even for patients with CS and R1/R2 resections, the OS is significantly better than that of non-CS patients. Furthermore, the value of CEA serum level appears to have a prognostic value for CS with R0 resection. Although a few concerns can be raised, the paper can potentially interest journal readers and add value to the current literature.
Major concerns:
The cohort includes a mixed group of patients with metastatic disease (solid organ metastases, peritoneal and lymph node metastases). This might influence the results.
Patients with technically resectable metastasis at initial diagnosis were excluded from the present cohort. However, in patients with gastric cancer and technically resectable metastases, what is the authors' attitude: surgery first or chemotherapy followed by surgery? Data about recurrence and survival in this group of patients should be provided.
The percentage of patients with CS in the present study is quite good. However, the number of patients with CS and Ro0 resection is not so high for the analyses. This should be stated as a limitation of the study.
How do the authors define induction chemotherapy?
The chemotherapy regimens were the same for the whole cohort? If not, are there any differences in chemotherapy types/ doses, timing, and duration between the CS and non-CS groups of patients? This might influence the results.
Why did the authors include as curative intent surgery also patients with D1 lymph nodes dissection? What were the criteria for performing either D1 or D2 lymph node dissection?
How do the authors assess the potential for an R0 resection preoperatively? It is unclear how the selection for CS was made, based only on preoperative or intraoperative data. Please clarify.
Patients with EGJ cancers probably should be excluded from the analyses.
Why did the authors include in the comparative analyses the neutrophil-to-lymphocyte ratio?
Define distant metastatic sites as hematogenous and lymphogenous. Define non-curable factors.
In Table 2, it is shown that a large part of patients with CS underwent only gastrectomy. One might find this surprising in the context of metastatic disease. Please explain. Furthermore, how the diagnosis of metastatic disease was made?
It is unclear how the analyses for early predictors for CS with R0 resection were made. Exclusion criteria should be explained for the analyses of only 156 patients out of 211. This might influence the results.
The authors should provide how to use the present study's data for clinical decision-making.
Comments on the Quality of English LanguageMinor editing of English language required.
Reviewer 3 Report
Comments and Suggestions for Authors
the article of Dr. Nakanishi et al. presents original research regarding the results of conversion surgery with R0 resection in metastatic gastric cancer. The subject is of novelty, and the study brings valuable information on the topic. Generally, the paper is well design and clearly written.
Some minor English language editing is required.
The Legend of the Figures should be more comprehensive
Comments on the Quality of English LanguageSome minor English language editing is required.